# Auxiliary sequential deposition enables 19%-efficiency organic solar cells processed from halogen-free solvents

Siwei Luo[1,10], Chao Li[1,2,10], Jianquan Zhang[3,10], Xinhui Zou[1,4,10], Heng Zhao[5], Kan Ding[6], Hui Huang[7], Jiali Song[2], Jicheng Yi[1], Han Yu [1], Kam Sing Wong [4], Guangye Zhang [7], Harald Ade [6], Wei Ma[5], Huawei Hu [8], Yanming Sun [2] ✉ & He Yan [1,9] ✉

High-efficiency organic solar cells are often achieved using toxic halogenated solvents and additives that are constrained in organic solar cells industry. Therefore, it is important to develop materials or processing methods that enabled highly efficient organic solar cells processed by halogen free solvents. In this paper, we report an innovative processing method named auxiliary sequential deposition that enables 19%-efficiency organic solar cells processed by halogen free solvents. Our auxiliary sequential deposition method is different from the conventional blend casting or sequential deposition methods in that it involves an additional casting of dithieno[3,2-b:2′,3′-d]thiophene between the sequential depositions of the donor (D18-Cl) and acceptor (L8-BO) layers. The auxiliary sequential deposition method enables dramatic performance enhancement from 15% to over 18% compared to the blend casting and sequential deposition methods. Furthermore, by incorporating a branched-chain-engineered acceptor called L8-BO-X, device performance can be boosted to over 19% due to increased intermolecular packing, representing top-tier values for green-solvent processed organic solar cells. Comprehensive morphological and time-resolved characterizations reveal that the superior blend morphology achieved through the auxiliary sequential deposition method promotes charge generation while simultaneously suppressing charge recombination. This research underscores the potential of the auxiliary sequential deposition method for fabricating highly efficient organic solar cells using environmentally friendly solvents.

In recent years, organic solar cells (OSCs) have rapidly developed due to their advantageous solvent processing and convenient production of large-area flexible devices by roll-to-roll techniques, which has led to unprecedented industrial prospects[1-5]. In particular, the emergence of non-fullerene acceptors (NFA) has revolutionized the development of OSCs, resulting in progressively improved power conversion efficiencies (PCE) beyond 19%[6-10]. Many of these achievements have been made through the regulation of donor-acceptor (D-A) interactions by controlling the bulk heterojunction (BHJ) morphology, where halogenated solvents such as chlorobenzene and chloroform, and solvent additives like 1,8-Diiodooctane (DIO) and 1-chloronaphthalene (CN), are widely used[11-13]. However, the massive utilization of such halogenated solvents during device fabrication could endanger human health, damage the

ecological environment, and increase economic costs, posing a potential barrier to the commercialization of OSCs[14,15].

Therefore, developing high-performance OSCs using halogen-free solvent processing has become a research focus in the field[16–22]. However, compared to halogenated solvents, the solubility of photoactive materials in halogen-free solvents is lower, negatively impacting the active layer morphology. Additionally, the relatively high boiling points of common halogen-free solvents, such as toluene, xylene, and trimethylbenzene, may induce over-aggregation of the donor and acceptor, resulting in inferior phase separation and damaging the photovoltaic performance of the device. Furthermore, halogen-free solvents not only affect the BHJ morphology from blend casting (BC), but also negatively influence the pseudo-planar heterojunction (PPHJ) from sequential deposition (SD), where the donor and acceptor materials are cast independently[23–26]. The challenge is that the second layer materials may not sufficiently infiltrate the first layer to form ideal vertical phase separation, leading to inadequate D−A interfaces and poor charge dissociation, and severe charge recombination. This reduces the open-circuit voltage ($V_{OC}$), short-circuit current density ($J_{SC}$), and fill factor (FF) of the devices, making it essential to optimize the morphology of active layers processed from halogen-free solvents[27–32].

Herein, we present an auxiliary sequential deposition (ASD) method using dithieno[3,2-b:2′,3′-d]thiophene (DTT) as a sequential processing auxiliary (SPA) to fabricate high-performance OSCs from halogen-free solvents. This strategy involves the sequential deposition of the D18-Cl donor polymer, SPA, and the NFAs from their toluene solutions (Fig. 1). In the ASD processing, D18-Cl is spin-coated from the hot solution, followed by the deposition of DTT from the cool solution. The highly crystalline DTT can penetrate the fiber network of D18-Cl and induce the preformation of a nano-porous polymer network. When the acceptor layer is processed, the NFAs can replace the DTT molecules, interdigitate with the D18-Cl fiber network, and generate highly crystalline domains in the PPHJ. Compared to traditional BC and SD processes, the ASD method enables better molecular packing, higher domain purity, and more desirable vertical phase separation of the active layers from halogen-free solvents. This is beneficial for suppressing charge recombination, as revealed by multiple steady-state and time-resolved techniques. Consequently, the D18-Cl:L8-BO devices based on the ASD method show significantly higher PCE of 18.03% than those of the BC (16.03%) and SD (15.50%) devices. Notably, changing the branch sites of the outer side chains on L8-BO from β to γ positions results in a stronger aggregation property and longer singlet exciton lifetime in the resulting NFA named L8-BO-X, further reducing charge recombination in the PPHJ. The ASD processing of D18-Cl:L8-BO-X achieves an even higher PCE of 19.04%, making it the top-ranking performance of OSCs from halogen-free solvents, and among the best results of layer-by-layer OSCs.

## Results
### Material design and properties
L8-BO is a high-performance NFA derived from Y6, owing to the delicate branched-chain engineering that has been proven to be an effective approach to fine-tuning the blend film morphology[33–39]. Here, we further move the branch sites of L8-BO (β-branch) by one more carbon to prepare an innovative NFA named L8-BO-X (γ-branch). The cyclic voltammetry (Supplementary Fig. 1) experiments show that changing the branch sites does not distinctly alter the energy levels of L8-BO (HOMO/LUMO: −5.75/−3.91 eV) and L8-BO-X (HOMO/LUMO: −5.74/−3.89 eV) as shown in Fig. 2a. Regarding the photon-harvesting properties, the two NFAs exhibit mostly the same absorption curves in the dilute solution state (Supplementary Fig. 2). Interestingly, in the solid state, L8-BO-X show slightly red-shifted absorption by 10 nm relative to L8-BO (806 vs 796 nm, Fig. 2b and Supplementary Table 1). In addition, the intensity ratio between the 0−0 and 0−1 vibrational transitions ($I_{0-0}/I_{0-1}$) are 1.47 and 1.56 for L8-BO and L8-BO-X, respectively. The higher $I_{0-0}/I_{0-1}$ of L8-BO-X suggest more J-aggregate characteristics in the solid state. The time-resolved photoluminescence (TRPL) spectra in Fig. 2c demonstrate that L8-BO-X has a longer exciton lifetime of 0.88 ns than L8-BO (0.59 ns), which echoes with the red-shifted absorption caused by the formation of more J-aggregated states and should facilitate exciton generation[40–42].

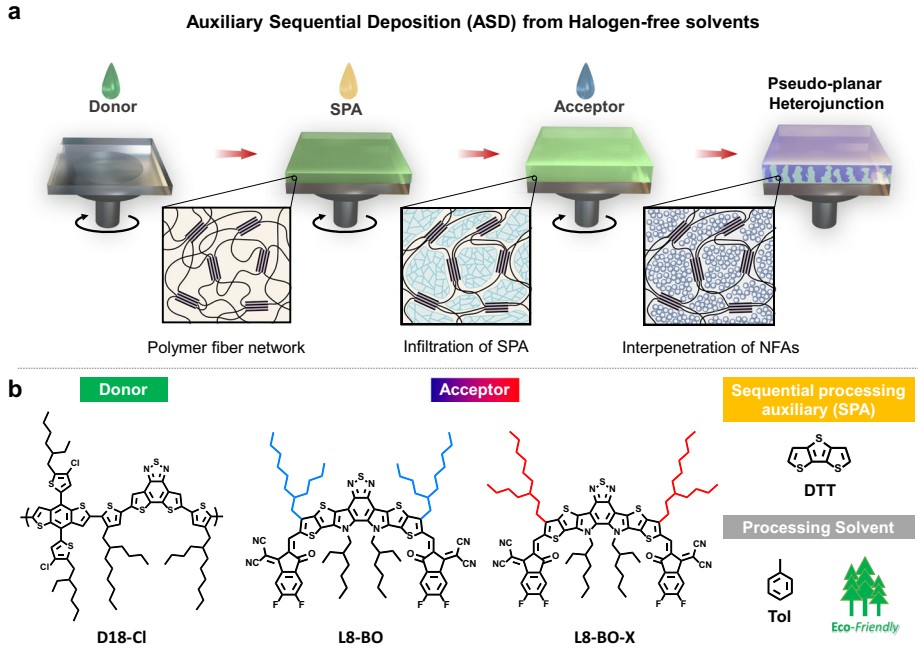

**Fig. 1 | Proposed auxiliary sequential deposition (ASD) processing using halogen-free solvents for OSCs. a** Schematic illustrations of the ASD device fabrication and possible nanoscale phase separation during device processing. **b** Chemical structure of the photoactive materials, D18-Cl, L8-BO, and L8-BO-X; DTT acts as the sequential processing auxiliary and toluene is used as halogen-free processing solvents.

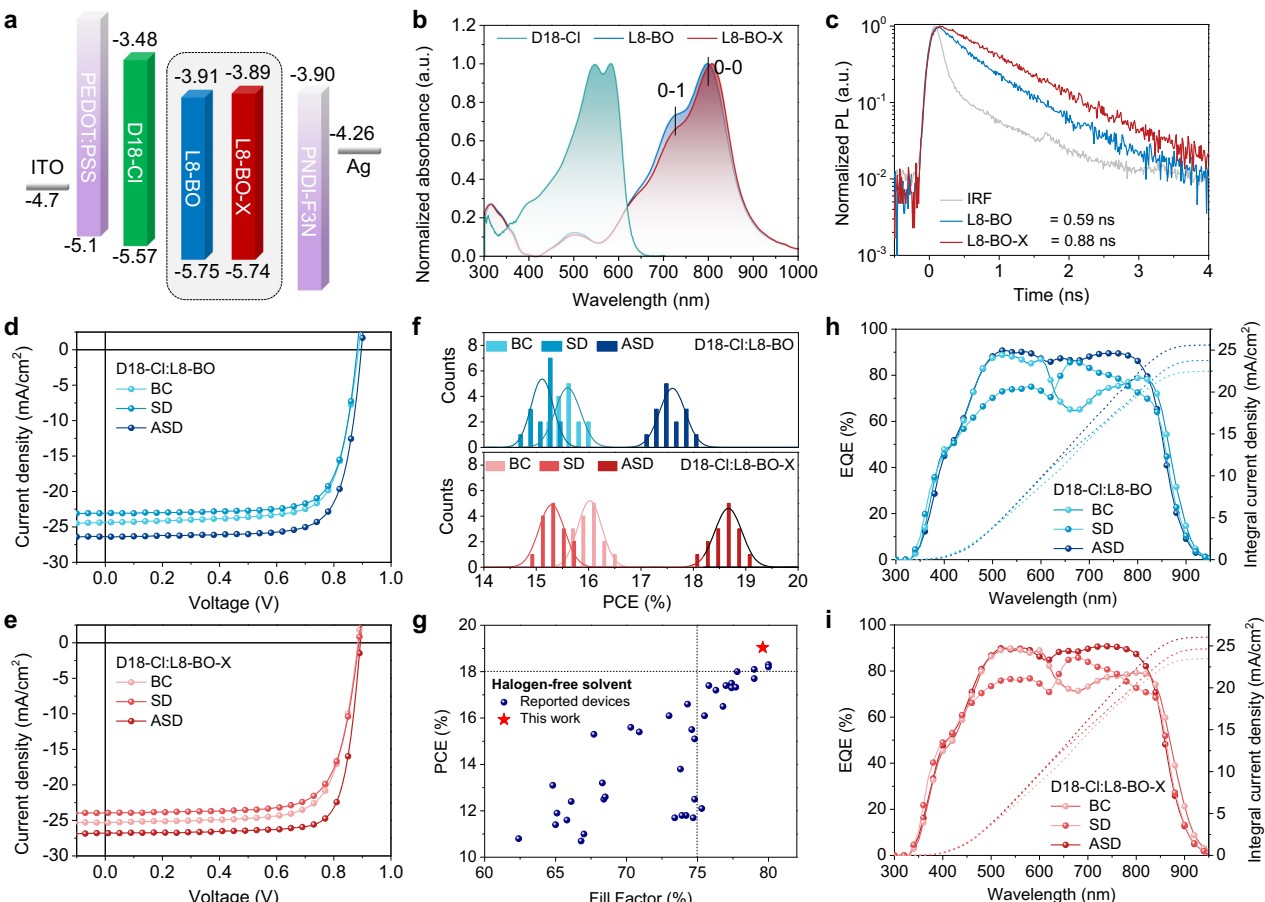

**Fig. 2 | Characterizations of materials and photovoltaic devices. a** Energy level alignment of the materials. **b** Absorption spectra of D18-Cl, L8-BO, and L8-BO-X in thin-film state. **c** Singlet exciton lifetime of L8-BO and L8-BO-X. The excitation and probe wavelengths are 780 and 860 nm, respectively; *J–V* curves of (**d**). D18-Cl:L8-BO and **e.** D18-Cl:L8-BO-X from the BC, SD, and ASD methods; **f.** Histogram of the PCE measurements for 15 devices. **g** Summary of reported devices from halogen-free solvents in the literature; EQE spectra (solid lines) and integrated *J*$_{SC}$ (dashed lines) of (**h**). D18-Cl:L8-BO and (**i**). D18-Cl:L8-BO-X from BC, SD, and ASD methods.

## Comparisons of photovoltaic devices from BC, SD, and ASD processing

A conventional device structure of ITO/PEDOT:PSS/active layer/ PNDI-F3N/Ag was fabricated to investigate the photovoltaic performance of the devices from the BC, SD, and ASD methods. In the BC method, the D:A mixture (1:1.5 by weight, 12.5 mg mL$^{-1}$ in toluene) was spin-cast to form photoactive layers on top of the substrate, while in the SD method, the D and A layers were sequentially deposited from toluene solutions with the concentrations of 7 and 10 mg mL$^{-1}$, respectively. The proposed ASD method involves one additional step of spin-casting the DTT molecules as the SPA material (20 mg mL$^{-1}$) between the deposition of the donor and acceptor layers. The UV-Vis spectra of the blend films are depicted in Supplementary Fig. 3. Noted that no solvent additives or post-treatment are used during active layer fabrication. The device optimizations are detailed in the Supporting Information (Supplementary Figs. 4–8 and Supplementary Tables 2–6). Figure 2d, e displays the typical current density versus voltage (*J–V*) characteristic curves, and Table 1 outlines the statistical photovoltaic parameters of the devices from the BC, SD, and ASD processing.

The BC devices of D18-Cl:L8-BO achieve a moderate PCE of 16.03% with a *V*$_{OC}$ of 0.886 V, a *J*$_{SC}$ of 24.38 mA cm$^{-2}$, and an FF of 74.2%. In comparison, the SD devices show an enhanced FF of 76.2% but a decreased *J*$_{SC}$ of 23.09 mA cm$^{-2}$, leading to an undesired PCE of 15.50%. In contrast, the ASD devices exhibit comprehensive improvements with a *V*$_{OC}$ of 0.896 V, a *J*$_{SC}$ of 26.37 mA cm$^{-2}$, and an FF of 76.3%, which

yields a superior PCE of 18.03%. Furthermore, the device results of D18-Cl:L8-BO-X also have the same trend but higher PCE of 16.50%, 15.73%, and 19.04% from the BC, SD, and ASD process, respectively, compared with D18-Cl:L8-BO. The enhancements from L8-BO to L8-BO-X mainly lie in the higher *J*$_{SC}$ for all three processing methods. Interestingly, the FF of D18-Cl:L8-BO-X (~74%) are lower than D18-Cl:L8-BO in the BC and SD devices, but are significantly increased to ~80% in the ASD devices, which suggests that the morphology of L8-BO-X is optimized in the ASD process but not in the BC and SD process. Figure 2f illustrates the PCE distributions of the devices from three processing methods, which obviously shows that the ASD process excels the BC and SD ones for both L8-BO and L8-BO-X-based OSCs. It should be noted that the 19.04% efficiency is among the top-ranking performance of the OSC devices from halogen-free solvents process (Fig. 2g). In addition, the ASD device of D18-Cl:L8-BO-X shows improved photostability than the BC and SD devices (Supplementary Fig. 9) under the maximum power point (MPP) tracking mode within a period of over 300 h, demonstrating the great potential of the ASD method in developing stable OSCs.

To verify the processing adaptability of ASD method, devices under ambient conditions and large-area devices were prepared. The air-processed devices show exemplary efficiency of over 17% (Supplementary Fig. 10, Supplementary Table 7), and the 1 cm² large-area devices show a satisfactory efficiency of over 15% (Supplementary Fig. 11, Supplementary Table 8), indicating that the ASD method has good processing adaptability, so that is more conducive to commercial

**Table 1 | Photovoltaic parameters of the D18-Cl:L8-BO and D18-Cl:L8-BO-X devices under AM1.5 G, 100 mW/cm$^2$**

| Method | Device | $V_{OC}$ (V) | $J_{SC}$ (mA/cm$^2$) | FF (%) | PCE (%) | $J_{SC,EQE}$ (mA/cm$^2$) | $\mu_h/\mu_e$ (10$^{-4}$ cm$^2$V$^{-1}$ s$^{-1}$) |
|---|---|---|---|---|---|---|---|
| BC (D:A) | D18-Cl:L8-BO | 0.886 (0.886 ± 0.001) | 24.38 (24.28 ± 0.29) | 74.2 (72.5 ± 1.5) | 16.03 (15.60 ± 0.24) | 23.77 | 7.87/5.29 |
| | D18-Cl:L8-BO-X | 0.885 (0.883 ± 0.002) | 25.31 (24.66 ± 0.47) | 73.7 (73.6 ± 1.4) | 16.50 (16.02 ± 0.22) | 24.63 | 7.92/5.38 |
| SD (D/A) | D18-Cl:L8-BO | 0.881 (0.881 ± 0.001) | 23.09 (23.02 ± 0.20) | 76.2 (74.5 ± 1.3) | 15.50 (15.11 ± 0.20) | 22.48 | 5.12/4.10 |
| | D18-Cl:L8-BO-X | 0.887 (0.887 ± 0.002) | 23.91 (23.62 ± 0.38) | 74.1 (73.0 ± 1.6) | 15.73 (15.29 ± 0.24) | 23.47 | 5.53/4.28 |
| ASD (D/SPA/A) | D18-Cl:L8-BO | 0.896 (0.896 ± 0.001) | 26.37 (26.36 ± 0.25) | 76.3 (74.5 ± 1.2) | 18.03 (17.60 ± 0.24) | 25.59 | 8.69/7.37 |
| | D18-Cl:L8-BO-X | 0.893 (0.891 ± 0.002) | 26.78 (26.67 ± 0.30) | 79.6 (78.5 ± 1.1) | 19.04 (18.67 ± 0.25) | 26.04 | 9.28/8.88 |

The data in parentheses are statistical results from 15 independent devices.

large-scale production. Further, to investigate the universality of ASD, non-fullerene (BTP-eC9, N3, FCC-Cl, and IDIC) based BC and ASD systems were fabricated (Supplementary Fig. 12). It is found that the ASD method shows excellent universality and comprehensive improvements over the BC method (Supplementary Table 9).

To understand the differences in $J_{SC}$ of these devices from various processing methods, external quantum efficiency (EQE) spectra of D18-Cl:L8-BO and D18-Cl:L8-BO-X are measured and plotted in Fig. 2h and i. By comparing the SD and ASD devices of the L8-BO and L8-BO-X-based devices, the improved $J_{SC}$ of the ASD devices originates from the higher EQE response (>85%) of both the donor and acceptor parts, implying that the conventional SD process does not form a suitable blend film morphology. Besides, although the BC devices achieve comparable EQE responses to the ASD ones in the donor part (400–600 nm), the responses in the acceptor part (600–800 nm) are substantially decreased, which also suggests inefficient charge transfer, dissociation, and transport. In addition, L8-BO-X shows a slightly higher EQE response than L8-BO in the BC, SD, and ASD devices, indicative of more efficient charge processes of L8-BO-X. Consequently, the integrated $J_{SC}$ values of the BC, SD, and ASD devices are determined to be 23.77, 22.48, and 25.59 mA cm$^{-2}$ for D18-Cl:L8-BO, and 24.63, 23.47, and 26.04 mA cm$^{-2}$ for D18-Cl:L8-BO-X, respectively, which is in good agreement with the J–V characteristics.

**Probing the blend film morphology of different processing methods**

The influence on the morphology from different processing methods is the key to understanding the variations in device performance. First, grazing-incidence wide-angle X-ray scattering (GIWAXS) analysis was performed to study the molecular stacking of neat and blend films[43]. The 2D diffraction patterns and 1D line-cut profiles are shown in Fig. 3a and b, respectively. Pristine L8-BO and L8-BO-X show similar π–π stacking, where the (010) peaks in the out-of-plane (OOP) direction are located at $q_z = 1.80$ Å$^{-1}$ and 1.79 Å$^{-1}$ with the π–π stacking distance ($d_{\pi-\pi}$) of 3.49 and 3.51 Å, and coherence length (CL) of 3.0 and 2.8 nm, respectively. However, L8-BO-X exhibits sharp peaks in the in-plane (IP) direction, especially for the one at $q_{xy} = 0.43$ Å$^{-1}$ showing a CL of 23.8 nm. This is much larger than that of L8-BO (4.0 nm), corresponding to the (11$\bar{1}$) peaks as reported in the literature[34], which suggests L8-BO-X has better long-range order than L8-BO. This is consistent with the electron mobility ($\mu_e$) measured by the space-charge-limited current (SCLC, Supplementary Fig. 13) method, where L8-BO and L8-BO-X have $\mu_e$ of $6.09 \times 10^{-4}$ and $8.23 \times 10^{-4}$ cm$^2$ V$^{-1}$ s$^{-1}$, respectively. When comparing the GIWAXS patterns of the BC and ASD films, the peaks at $q_{xy} = 0.42$ Å$^{-1}$ of D18-Cl:L8-BO (CL = 8.1 nm) and 0.38 Å$^{-1}$ of D18-Cl:L8-BO-X (CL = 37.1 nm) are much more obvious in the ASD films, indicating stronger lamellar stacking. In addition, the ASD film of D18-Cl:L8-BO-X shows a very sharp peak at $q_z = 1.74$ Å$^{-1}$ with a much larger CL of 8.9 nm than the other blend films (CL - 2.6–2.9 nm). These GIWAXS data thus prove that the ASD process yields highly crystalline domains relative to the BC process, especially for the L8-BO-X-based blend films.

Then, atomic force microscopy (AFM, Fig. 3c) images of all the blend films show easily discernible interpenetrated fiber networks, which should arise from the strong aggregation of the D18-Cl polymer[44]. The BC films exhibit certain hollow space and unevenly distributed surface morphology with large root-mean-square (RMS) roughness of 2.06 and 1.45 nm for D18-Cl:L8-BO and D18-Cl:L8-BO-X, respectively. In contrast, the RMS roughness of the SD (1.17 and 0.99 nm) and ASD (0.86 and 0.89 nm) films are significantly reduced for D18-Cl:L8-BO and D18-Cl:L8-BO-X, respectively, forming fine and subtle fiber network structures. Therefore, both sequential processing produces the more desired surface morphology, especially for the ASD method. Furthermore, Fig. 3d depicts the resonant soft X-ray scattering (RSoXS) profiles of the blend films[45]. Whilst all the blend films have similar domain spacing, the two ASD films have higher integrated scattering intensity (ISI) than the corresponding BC and SD films. The domain purity of the BC, SD, and ASD films are further determined to be 0.94, 0.81, and 0.97 for D18-Cl:L8-BO, and 0.90, 0.85 and 1.00 for D18-Cl:L8-BO-X, respectively, which trends in line with the hole mobility ($\mu_h$) and $\mu_e$ as shown in Supplementary Figs. 14, 15 and tabulated in Table 1 and Supplementary Table 10. The morphology characterizations of GIWAXS, AFM, and RSoXS collectively demonstrate that the ASD method facilitates desirable phase separation to form highly crystalline fiber networks with high phase purity, which enhance charge transport and suppress charge recombination.

Subsequently, film-depth-dependent light absorption spectroscopy (FLAS) was performed to probe the exciton generation in the vertical direction of the photoactive layers[46–48]. The FLAS characteristics of the BC, SD, and ASD films of D18-Cl:L8-BO-X are shown in Fig. 4a–c, respectively (the data of D18-Cl:L8-BO are shown in Supplementary Fig. 16). Different film processing methods have varying degrees of influence on the absorption peak shifts of L8-BO-X at around 800 nm as film depth changes. L8-BO-X from the ASD film has a minimal peak shift among the three films, suggesting that the LUMO level of L8-BO-X from the ASD film is mostly invariable at different film depths[49]. The minimal spatial fluctuation of energetic levels can effectively avoid the formation of the low-energy localized states as traps along the charge transporting pathways, which is beneficial for improving electron transporting and FF of the ASD film[50]. Furthermore, the exciton generation contours are numerically simulated using the transfer matrix method from the FLAS profiles (Fig. 4d–f), and the exciton generation rate (G) of each blend film can then be obtained (Fig. 4g–i)[51–53]. From the contour figures, it is obvious that the exciton generation from the photoactive layers mostly occur near the bottom part (near PEDOT:PSS/ITO) for both BC and SD films. Besides, the maximum G values are located at 66 and 82 nm of film depth for BC and SD films, respectively. Upon exciton dissociation into free charges, the generated electrons need to cross the distance of tens of nanometers to be collected by the cathode, which dramatically increases the risk of charge recombination[8, 54]. Surprisingly, the charges of the ASD film are generated at the middle region of the photoactive layer, with the maximum G values located at 56 nm of film depth. Therefore, both holes and electrons can more effectively transport across the

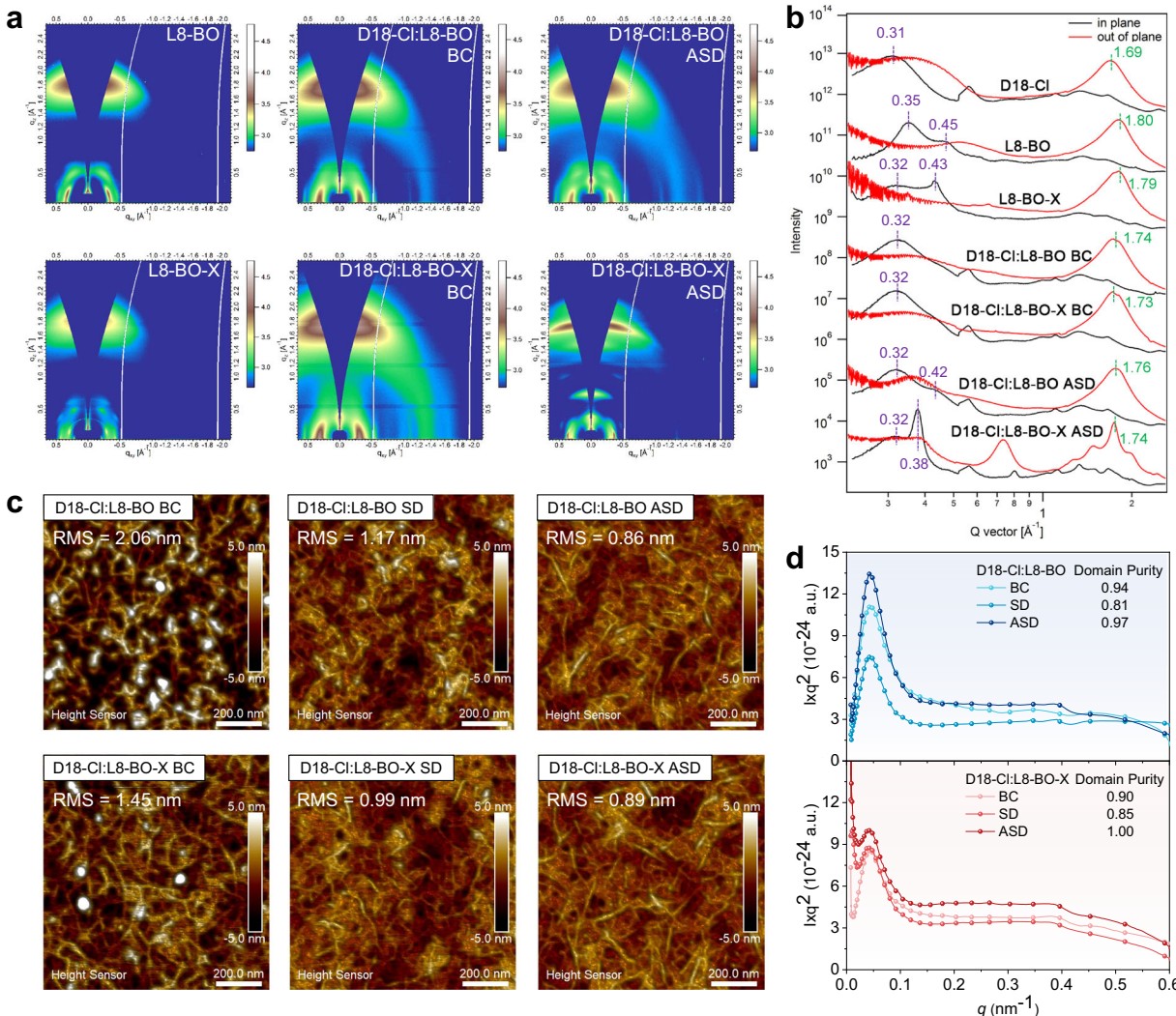

**Fig. 3 | Thin-film morphology from different processing methods. a** 2D GIWAXS patterns of the D18-Cl neat film, ASD D18-Cl film, L8-BO neat film, L8-BO-X neat film and D18-Cl:L8-BO, D18-Cl:L8-BO-X BC and ASD blend films. **b** In-plane (black) and out-of-plane (red) 10° sector averaged GIWAXS profiles of the corresponding 2D patterns. **c** AFM height images and (**d**). RSoXS profiles of the D18-Cl:L8-BO and D18-Cl:L8-BO-X films from three processing methods.

photoactive layer, which is conducive to the higher photocurrent and FF of the ASD films.

## Understanding of photophysical processes

The charge recombination mechanism is investigated by correlating the illumination intensity ($P_{light}$) and $V_{OC}/J_{SC}$ of the devices. According to the formula $V_{OC} = nkT \ln(P_{light})/q$, Fig. 5a shows that D18-Cl:L8-BO from different processing methods exhibit similar slopes of ~1.14 $kT/q$. However, for D18-Cl:L8-BO-X, the ASD device shows an obviously smaller slope of 1.04 $kT/q$ than the BC (1.12 $kT/q$) and SD (1.18 $kT/q$) device, indicative of the effectively prohibited trap-assisted recombination. According to the formula $J_{SC} \propto P_{light}^{\alpha}$, Fig. 5b illustrates that the α values of the BC and ASD devices based on L8-BO and L8-BO-X are larger than those of the SD devices and very close to unity, signifying that the bimolecular recombination is suppressed. Therefore, the ASD method is superior to the conventional BC and SD methods, especially for L8-BO-X. To study the exciton dissociation of the devices from different processing methods, photocurrent density ($J_{ph}$) versus effective voltage ($V_{eff}$) curves are plotted in Fig. 5c. For both L8-BO and L8-BO-X-based devices, the saturated $J_{ph}$ ($J_{sat}$) of the SD films are significantly reduced compared with those of the BC and ASD films, demonstrating that the exciton generation of the SD devices is insufficient. On the one hand, despite the comparable $J_{sat}$ to the ASD devices, the BC devices show lower exciton dissociation efficiencies ($\eta_{diss} = 91.78\%$ and 92.37%) than the ASD devices ($\eta_{diss} = 96.57\%$ and 96.84%) for both L8-BO and L8-BO-X-based blends, respectively, accounting for their relatively inferior $J_{SC}$ values. On the other hand, all the L8-BO-X-based devices have higher $\eta_{diss}$ values than the L8-BO-based counterparts, which demonstrates that L8-BO-X has excellent charge dissociation properties. As a result, the highest $\eta_{diss}$ value of D18-Cl:L8-BO-X from the ASD method thus partially explains the enhancement in the $J_{SC}$ of the corresponding devices.

Apart from steady-state experiments, time-resolved techniques were also employed to further understand the charge dynamics. First of all, TRPL spectra of each blend film were recorded and plotted in Fig. 5d. For both L8-BO and L8-BO-X-based systems, the exciton lifetimes ($\tau_{ex}$) are shorter in the films processed from the BC (0.281 ns and 0.286 ns) and ASD (0.250 ns and 0.287 ns) method, implying that the charge transfer process in these films is more efficient than those in the SD films (0.289 ns and 0.330 ns). Subsequently, transient photovoltage (TPV) and transient photocurrent (TPC) measurements were conducted for all devices. The charge carrier lifetimes ($\tau_{TPV}$) from TPV measurements (Fig. 5e) show the sequence of ASD (1.028 and 1.180 μs) > SD (0.876 and 0.830 μs) > BC (0.429 and 0.390 μs) films for D18-Cl:L8-BO and

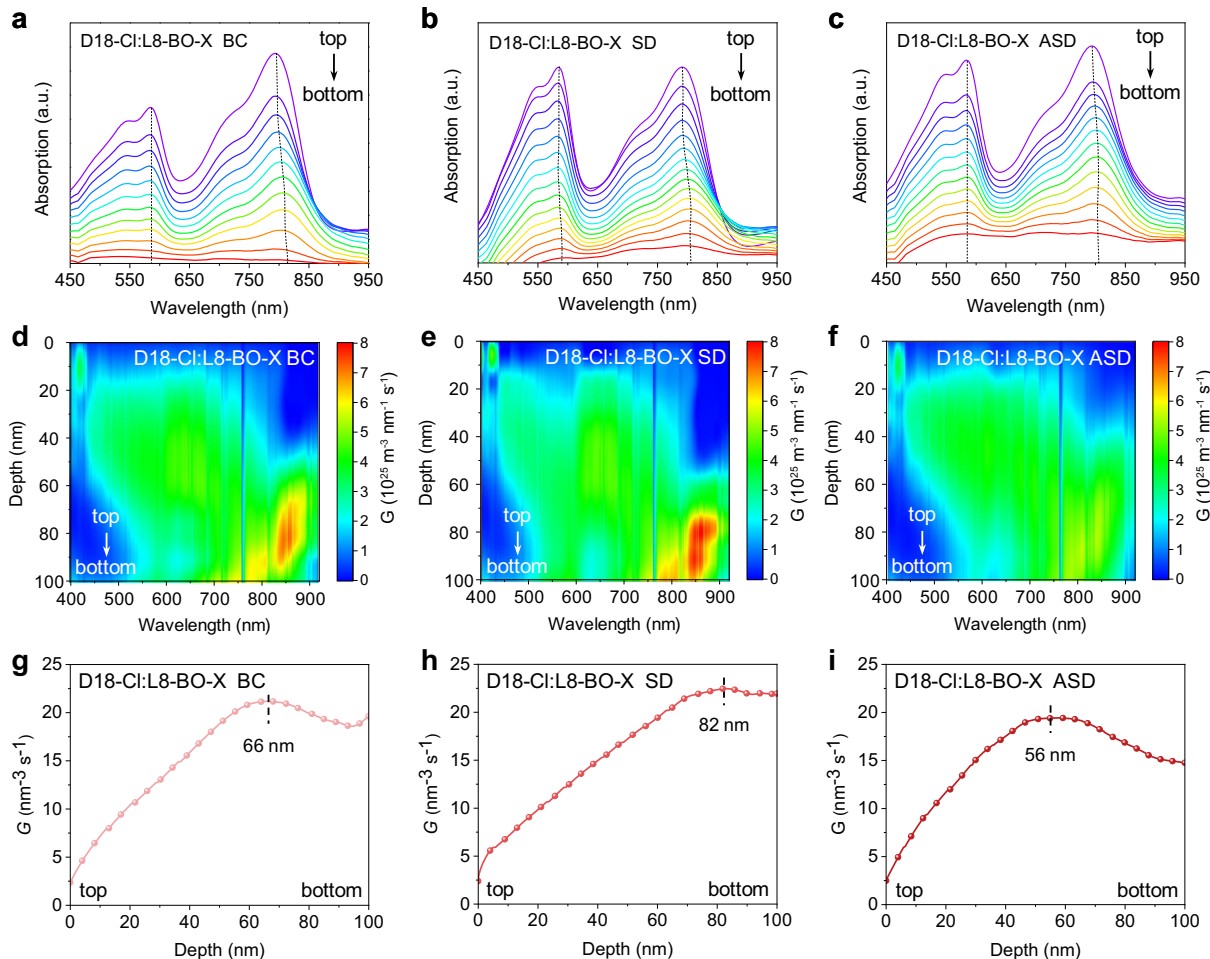

**Fig. 4 | Film-depth-dependent light absorption spectroscopy characterizations.** The FLAS characteristics of D18-Cl:L8-BO-X from (**a**) BC, (**b**) SD, and (**c**) ASD processing. Numerical simulations for the exciton generation contours of D18-Cl:L8-BO-X from (**d**) BC, (**e**) ASD, and (**f**) ASD processing, where the noise arise from the fluctuation of the AM 1.5 G solar spectrum. Dependence of the simulated exciton generation rate (G) on the film depth of D18-Cl:L8-BO-X from (**g**) BC, (**h**) SD, and (**i**) ASD processing.

D18-Cl:L8-BO-X, respectively. This means the ASD method can suppress charge recombination of the active layer, which is in line with the light-intensity-dependent $V_{OC}$/$J_{SC}$ experiment. Besides, Fig. 5f presents the TPC measurements and charge extraction times ($\tau_{TPC}$). For D18-Cl:L8-BO, the BC and ASD films show shorter $\tau_{TPC}$ of 0.139 and 0.149 μs, respectively, than the SD films (0.211 μs). Similarly, for D18-Cl:L8-BO-X, shorter $\tau_{TPC}$ of 0.128 and 0.144 μs were observed for the BC and ASD films, respectively, relative to that for the SD film (0.166 μs), indicating the faster charge sweep-out and larger charge extraction capability. All these device-physics-related characterizations demonstrate that the ASD processing method has multiple advantages over the SD one in terms of efficient charge transfer and dissociation, suppressed charge recombination, and faster charge extraction.

Femtosecond transient absorption spectroscopy (fs-TAS) was performed to study the charge generation and recombination process. The fs-TAS profiles of pure L8-BO and L8-BO-X excited at 800 nm are shown in Supplementary Fig. 17. Positive features peaked at ~830 nm for L8-BO and ~840 nm for L8-BO-X are assigned to the ground-state bleach (GSB), while the negative signal at ~920 nm is assigned to the excited-state absorption (ESA). L8-BO-X exhibits a longer decay time than L8-BO, which is consistent with the longer-lived TRPL and results in better potential in the non-radiative loss of the blend[8,30,51]. Upon selective excitation on the D18-Cl:L8-BO and D18-Cl:L8-BO-X at 800 nm, we see the rise of a positive $\Delta T/T$ signal at ~830 nm (GSB) and

a negative signal ~900 nm (ESA) at the early time stage (see Fig. 6a for D18-Cl:L8-BO-X ASD sample and Supplementary Figs. 18–20 for other samples). A positive signal at 600 nm that corresponds to the absorption of D18-Cl builds up at a later stage and reaches the maximum within 100 ps, which is ascribed to the hole transfer process from the acceptor to the donor. We fit the rise of donor GSB (Fig. 6b and c) with the biexponential model (Supplementary Table 11), where $\tau_1$ represents the dissociation time of the charge-transfer states at the donor/acceptor interfaces and $\tau_2$ represents the diffusion time of excitons to the D/A interfaces[55,56]. Both SD samples show significantly longer $\tau_1$ and $\tau_2$ than the BC samples, which are due to the longer dissociation time. On the other hand, the ASD films exhibit the fastest exciton dissociation and the shortest diffusion time. Furthermore, the GSB kinetics are the longer-lived for the ASD films as shown in Fig. 6d and e since the residue of the acceptor GSB for blends are due to the polarons, the longer-lived signal in the ASD devices further proves the suppressed recombination. Hence, the fs-TAS results match the above device and morphology characterizations, which explains that the ASD process enables faster exciton dissociation for superior $J_{SC}$ and shorter diffusion time for reduced charge recombination.

## Discussion

In summary, we have developed an ASD process as a versatile method to obtain optimized PPHJ structures for highly efficient OSCs from

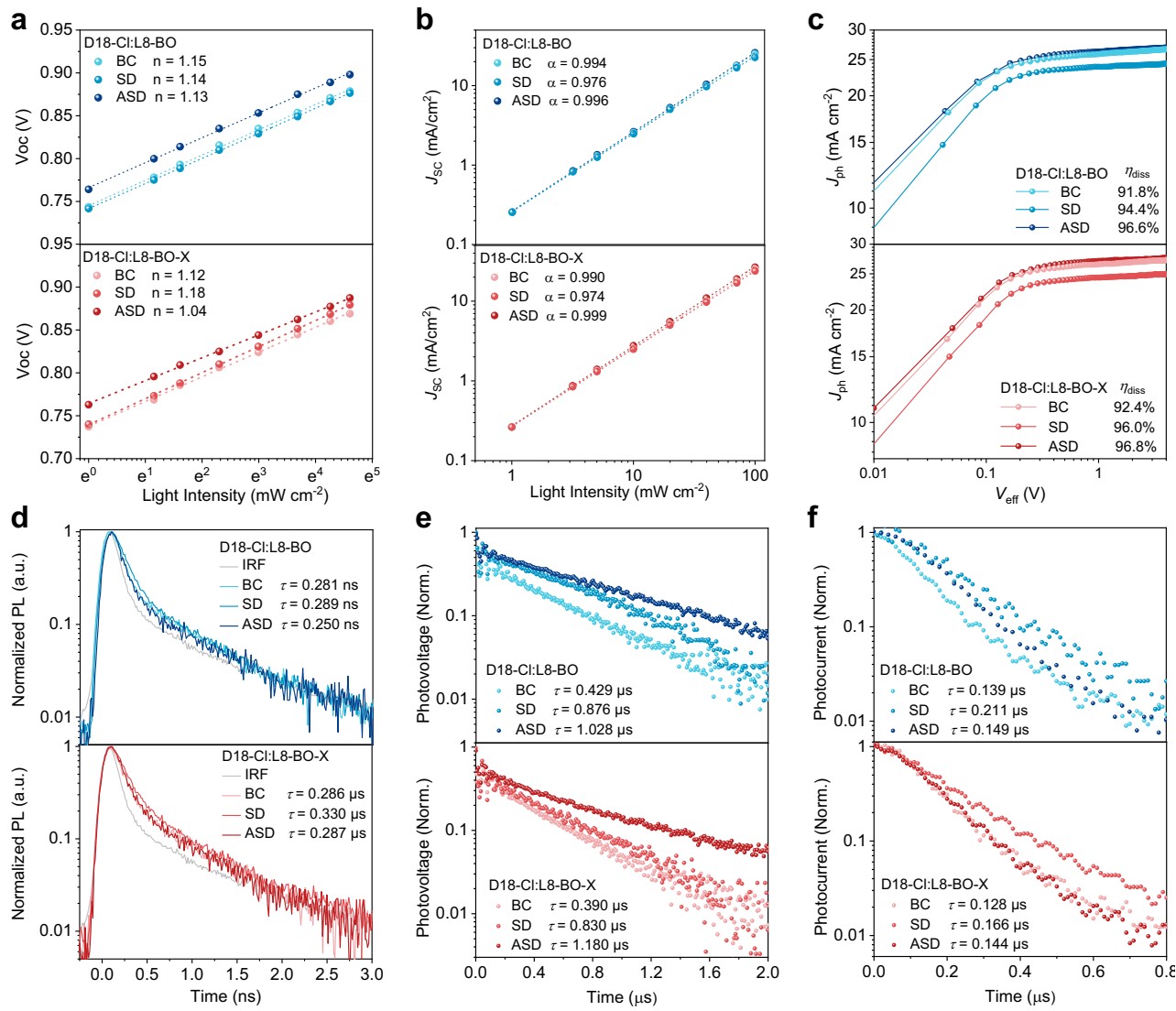

**Fig. 5 | Steady-state and time-resolved characterizations of photovoltaic devices. a** Light intensity dependence of $V_{OC}$. **b** Light intensity dependence of $J_{SC}$. **c** Photocurrent density ($J_{ph}$) as a function of the effective voltage ($V_{eff}$). **d** TRPL decay spectra. **e** Transient photovoltage. **f** Transient photocurrent of the D18-Cl:L8-BO (blue) and D18-Cl:L8-BO-X (red) devices processed from the BC, SD, and ASD methods.

halogen-free solvents. The traditional SD process using halogen-free solvents may result in undesirable phase separation because of low material solubility and thus poor interpenetration of the donor and acceptor. It is discovered that the treatment of DTT between the deposition of the D18-Cl and acceptor layers could induce better material crystallinity and phase separation owing to the preformation of nano-porous fiber networks. The casting of the acceptor onto the nano-porous D18-Cl film should result in better phase separation between the donor and acceptor, which significantly enlarges the D/A interfaces for efficient charge separation, and creates highly crystalline domains for improving charge transport and reducing charge recombination. As a result, the D18-Cl:L8-BO devices using toluene as the halogen-free solvent based on the ASD method show a dramatically enhanced PCE of 18.03% relative to those based on the BC (16.03%) and SD (15.50%) methods. Moreover, L8-BO-X from the branched-chain engineering of L8-BO yields an even higher performance of 19.04% based on the ASD method, mainly due to the stronger aggregation and longer exciton lifetime. This is among the best values for OSCs from halogen-free solvents. The ASD method developed in this work represents a breakthrough in OSCs, which potentially become a

general process toward the large-scale halogen-free solvent processing of high-performance solar cell devices.

## Methods

### Device fabrication and testing

The device's fabrication follows the classic method: ITO/PEDOT:PSS/ Active Layer/PNDI-F3N/Ag. Indium Tin Oxide (ITO) substrates were successively washed in ultrasonic cleaning agent, deionized water, acetone and isopropanol for 20 minutes, respectively. Then, the substrates were put into the oven to dry at 120 °C overnight. Before using, 30 min UV-Ozone pretreatment was carried out to improve fill factor. PEDOT:PSS (Heraeus Clevios P VPA 4083) was spined onto the top of ITO at 5000 rpm, then the substrates were put onto hotplate of 150 °C to anneal for 15 min thus optimize crystallinity and smoothness for further processes.

For BC Devices, the donor/acceptor ratio of the active layer solvent is of 1:1.5 with total concentration of 12.5 mg mL$^{-1}$ in toluene, For BC devices, the solvent should be stirred for 1 h at 110 °C to ensure complete dissolution. Then the active layer was spined onto the substrates at 3000 rpm. For SD and ASD devices, the donor and

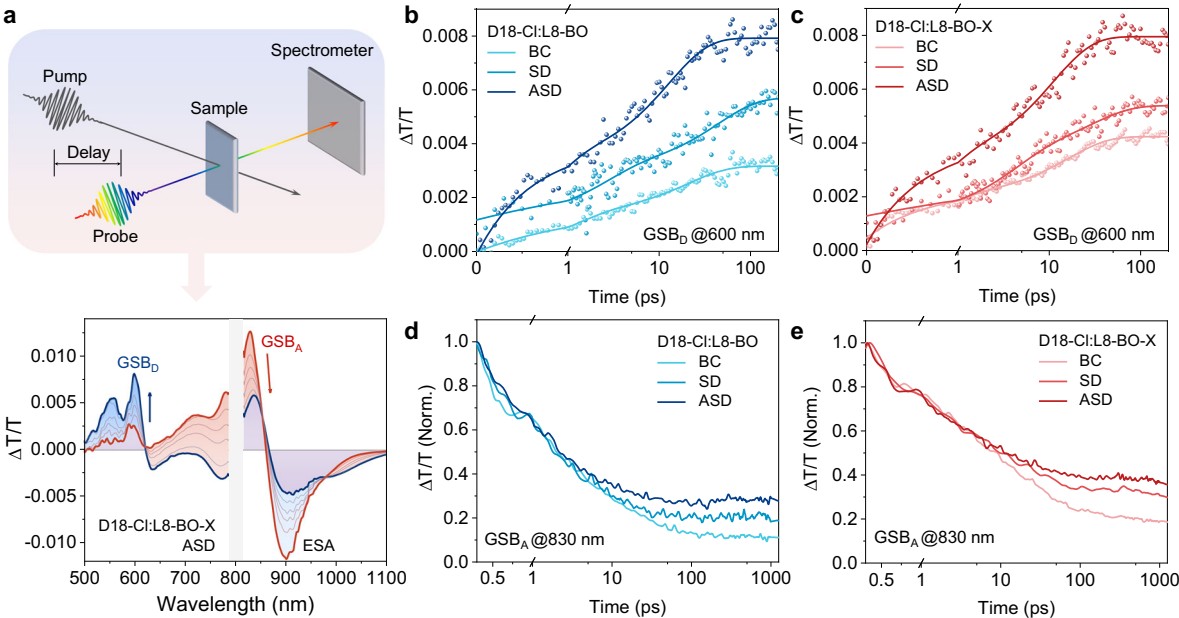

**Fig. 6 | Femtosecond transient absorption spectroscopy (fs-TAS) and study on charge recombination. a** Schematic illustration of the pump-probe technique for TAS measurements, and the femtosecond transient absorption spectroscopy (fs-TAS) excited at 800 nm shows several key features, including ground-state bleach of donor (GSB$_D$) and acceptor (GSB$_A$), excited-state absorption (ESA), and hole polarons. TAS profiles probed at 600 nm for (**b**) D18-Cl:L8-BO and (**c**) D18-Cl:L8-BO-X from three processing methods. TAS profiles probed at 830 nm for (**d**) D18-Cl:L8-BO and (**e**) D18-Cl:L8-BO-X from three processing methods.

the acceptor were dissolved in toluene with the concentrations of 7 and 10 mg mL⁻¹ and stirred for 1 h at 110 and 80 °C, respectively, to ensure complete dissolution. The SPA material DTT was dissolved in toluene with a concentration of 20 mg mL⁻¹. For SD devices, the donor was spined onto the substrates at 4000 rpm for 30 s. Then, the acceptor was spined onto the top of donor layer at 3000 rpm for 30 s. For ASD devices, after the donor was spined onto the substrates at 4000 rpm to form a uniform donor layer, the SPA material DTT was spined onto the top of the donor layer at 2000 rpm for 60 s and stood for 10 min to ensure the preformation of a nano-porous polymer structure. Then, the acceptor was spined onto the donor layer at 3000 rpm. After that, the electron transfer layer, PDI-F3N was spined onto the active layer at 3000 rpm (1.5 mg mL⁻¹ in methanol solution with 0.5 vol. % AcOH). Then, the substrate was transfer to metal evaporator (KC resistance-type thermal metal coating instrument) to evaporate 100 nm silver on the top of the device as cathode. Then the devices were encapsulated by UV light-cured resin and optical glass. After that, the thickness of the active layer measured by the Woollam Alpha-SE ellipsometer. The *J–V* curves of all encapsulated devices were covered by 5 mm² mask and measured using a Keithley 2400 Source Meter in air under AM 1.5 G (100 mW cm⁻²) using a Newport solar simulator. The light intensity was calibrated using a standard Si diode (with KG5 filter, purchased from PV Measurement to bring spectral mismatch to unity). The scanning range is from −0.5 V to 1.0 V. The scan speed and dwell times were fixed at 0.01 V/step and 20 ms, respectively.

### General information
Bruker AV-400 MHz NMR spectrometer was used to record ¹H and ¹³C NMR spectra on Chemical shifts are reported in parts per million (ppm, δ). ¹H NMR and ¹³C NMR spectra were referenced to tetramethylsilane (0 ppm) for CDCl₃. Mass spectra were collected on a MALDI Micro MX mass spectrometer, or an API QSTAR XL System.

### Optical characterizations
A Perkin Elmer Lambda 20 UV/VIS Spectrophotometer was used to obtain film UV-Vis absorption spectra. A solution of acceptors with concentration of 20 mg mL⁻¹ in chloroform was spin coated on optical glass substrates to create acceptor films. The blend films were created by spin-coated their chloroform solution (10 mg mL⁻¹) on optical glass substrates at 3000 rpm for 30 s. The solution UV–Vis absorption spectra were collected from the solution of three SMAs with the concentration of $1.0 \times 10^{-5}$ M in chloroform.

### Electrochemical characterizations
Cyclic voltammetry was carried out on A CHI610E electrochemical workstation with three-electrode configuration was used to process cyclic voltammetry test. We set Ag/AgCl to be the reference electrode and a Pt plate to be the counter electrode, and a glassy carbon as the working electrode. Tetrabutylammonium hexafluorophosphate in anhydrous acetonitrile with a concentration of 0.1 mol L⁻¹ was used as the supporting electrolyte. The solid films were drop-casted on the working electrode from a chloroform solution with a concentration of 5 mg mL⁻¹. Potentials were referenced to the ferrocenium/ferrocene couple by using ferrocene as external standards in acetonitrile solutions. With the scan rate of 0.05 V s⁻¹, the LUMO levels were calculated by $-(E_{re} - E_{fc} + 4.8)$ eV and the HOMO levels were calculated by $-(E_{ox} - E_{fc} + 4.8)$ eV.

### EQE measurements
Enlitech QE-S EQE system was used to measure EQEs. The EQE system was calibrated by standard Si diode. Monochromatic light was generated from a Newport 300 W xenon lamp.

### Device photostability
Photostability of the devices was performed by using Maximum Power Point (MPP) tracking mode (YH-VMPP-IV-16). The encapsulated devices were measured under continuous illumination by LED light source

(380–810 nm, one solar intensity) under the ambient air conditions (the relative humidity was 40 %, the tested temperature was 45 °C).

## SCLC measurements

The electron and hole mobilities were measured by using the method called space-charge limited current (SCLC) for electron-only and hole-only devices. The structure of electron-only devices was ITO/ZnO/active layer/PDINO/Al and the hole-only devices were fabricated with the structure of ITO/PEDOT:PSS/active layer/MoO$_3$/Ag. The charge carrier mobility was determined by fitting the dark current to the model of a single carrier SCLC according to the Mott−Gurney law: $J = 9\varepsilon_0\varepsilon_r\mu V^2/8L^3$, where $J$ is the measured current density, $L$ is the film thickness of the active layer, µ is the mobility of charge carrier, $\varepsilon_r$ is the relative dielectric constant of the transport medium component, and $\varepsilon_0$ is the permittivity of vacuum (8.85419 × 10$^{-12}$ C V$^{-1}$ m$^{-1}$), V is the difference of applied voltage ($V_{app}$) and offset voltage ($V_{BI}$). The mobility of charge carriers can be calculated from the slope of the $J^{1/2} \sim V$ curves.

## AFM characterizations

The AFM images were recorded using a Bruker multimode 8 AFM.

## Grazing incidence wide-angle x-ray scattering (GIWAXS) characterization

GIWAXS measurements were performed at beamline 7.3.3 at the Advanced Light Source. Samples were prepared on Si substrates using identical blend solutions as those used in devices. The 10 keV X-ray beam was incident at a grazing angle of 0.11°–0.15°, selected to maximize the scattering intensity from the samples. The scattered x-rays were detected using a Dectris Pilatus 2 M photon counting detector.

## Resonant soft x-ray scattering (RSoXS)

RSoXS transmission measurements were performed at beamline 11.0.1.2 at the Advanced Light Source (ALS). Samples were prepared on a PSS modified Si substrate under the same conditions as those used for device fabrication, and then transferred by floating in water to a 1.5 mm × 1.5 mm, 100 nm thick Si$_3$N$_4$ membrane supported by a 5 mm × 5 mm, 200 µm thick Si frame (Norcada Inc.). 2-D scattering patterns were collected on an in-vacuum CCD camera (Princeton Instrument PI-MTE). The sample detector distance was calibrated from diffraction peaks of a triblock copolymer poly(isoprene-b-styrene-b-2-vinyl pyridine), which has a known spacing of 391 Å. The beam size at the sample is approximately 100 µm by 200 µm.)

## Film-depth-dependent light absorption spectroscopy (FLAS)

Film-depth-dependent light absorption spectra were acquired by an in-situ spectrometer (PU100, Shaanxi Puguang Weishi Co. Ltd.) (Shaanxi, China) equipped with a soft plasma-ion source. The power supply for generating the soft ionic source was 100 W with an input oxygen pressure ~ 10 Pa. The film surface was incrementally etched by the soft ion source, without damage to the materials underneath the surface, which was in situ monitored by a spectrometer. From the evolution of the spectra and the Beer−Lambert's Law, film-depth-dependent absorption spectra were extracted. The exciton generation contour is numerically simulated upon inputting sub-layer absorption spectra into a modified optical transfer-matrix approach.

## Transient photocurrent (TPC) and transient photovoltage (TPV) measurement

The device was mounted on a conductive clip and under steady-state illumination from focused Quartz Tungsten-Halogen Lamp light source. The measurements were performed with the background response of open-circuit voltage. An optical perturbation is applied to the device with a 1 kHz femtosecond pulse laser. The TPV signal was acquired by a digital oscilloscope at open-circuit condition. TPC signal was measured under approximately short-circuit condition by applying a 50 Ω resistor.

## Time-resolved photoluminescence spectroscopy

Measurements were carried out on encapsulated thin films. A Ti:Sapphire oscillator (Coherent Mira 900) operating at 76 MHz repetition rate was applied to excite the sample. The PL was collected and guided into a spectrometer equipped with a silicon single-photon counter to carry out time correlated single photon counting (TCSPC), integrating majority of the PL.

## Transient absorption spectroscopy

Two beams were split from the 800 nm output of a Ti:Sapphire laser amplifier (Coherent Legend Elite, repetition rate of 1 kHz, 100 fs). One of the split outputs is used to photoexcite the sample either directly at 800 nm or at other wavelengths generated in an optical parametric amplifier (Coherent OPerA Solo). The other split output is used to generate a white light continuum probe pulse by focusing the beam onto a sapphire plate. The optical pulses were spatially overlapped in the encapsulated thin-film sample, and temporally delayed using a motor-driven delay stage. The probe was guided into a spectrometer (Acton instruments SpectraPro 275) with a 50 LP mm$^{-1}$ grating and detected by a Si array detector. The pump-induced change in transmission ($\Delta T/T$) was recorded shot by shot. The experiment is performed with pump fluence of ~3 µJ cm$^{-2}$ per pulse. Background signals were subtracted from the signal, and the chirp response is corrected. For the hole transfer study, a pump of 800 nm is used to selectively photoexcite the acceptor part. For the donor excitation, a pump of 550 nm is used to preferentially excite the donor.

## Reporting summary

Further information on research design is available in the Nature Portfolio Reporting Summary linked to this article.

## Data availability

The experiment data generated in this study are provided in the Supplementary Information/Source Data file. Source data are provided with this paper.

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

## Acknowledgements

S.L., C.L., J.Z. and X.Z. contributed equally to this work. This work was financially supported by the National Natural Science Foundation of China (Grant No. 51825301). H. Yan. appreciates the support from the National Key Research and Development Program of China (No. 2019YFA0705900) funded by MOST, the Basic and Applied Research Major Program of Guangdong Province (No.2019B030302007), National Natural Science Foundation of China (NSFC, No. 91433202, 22075057), the Shen Zhen Technology and Innovation Commission through (Shenzhen Fundamental Research Program, JCYJ20200109140801751), the Hong Kong Research Grants Council (research fellow scheme RFS2021-6S05, RIF project R6021-18, CRF project C6023-19G, GRF project 16310019, 16310020), Hong Kong Innovation and Technology Commission (ITC-CNERC14SC01) and Foshan-HKUST (Project NO. FSUST19-CAT0202). W.M. thanks for the support from National Natural Science Foundation of China (21704082, 21875182), Key Scientific and Technological Innovation Team Project of Shaanxi Province (2020TD-002), 111 project 2.0 (BP2018008). X-ray data were acquired at beamlines 7.3.3 at the Advanced Light Source, which is supported by the Director, Office of Science, Office of Basic Energy Sciences, of the U.S. Department of Energy under Contract No. DE-AC02-05CH11231. The authors thank Dr. Eric Schaible and Dr. Chenhui Zhu at beamline 7.3.3 for assistance with data acquisition.

## Author contributions

S.L. conceived the idea and fabricated the devices. C.L. designed and synthesized the L8-BO-X material. J.Z. characterized the devices and wrote the manuscript. X.Z. conducted the TRPL, TPV, TPC, and fs-TAS experiments under the supervision of K.S.W. The GIWAXS characterizations were conducted by H.Z. under the supervision of W.M. The RSoXS characterizations were conducted by K.D. under the supervision of H.A. The FLAS characterizations were measured by H.Huang under the supervision of G.Z. J.S. tested the device stability. J.Y. and H.Yu provided the D18-Cl polymer. H.Hu, Y.S., and H.Yan provided important insights during the project design and supervision.

## Competing interests

The authors declare no competing interests.

## Additional information

[1]Department of Chemistry, Guangdong-Hong Kong-Macao Joint Laboratory of Optoelectronic and Magnetic Functional Materials, Energy Institute and Hong Kong Branch of Chinese National, Engineering Research Center for Tissue Restoration and Reconstruction, Hong Kong University of Science and Technology, Clear Water Bay, Kowloon, Hong Kong 999077, China. [2]School of Chemistry, Beihang University, 100191 Beijing, China. [3]School of Science and Engineering, Shenzhen Institute of Aggregate Science and Technology, The Chinese University of Hong Kong, 518172 Shenzhen, Guangdong, China. [4]Department of Physics, Hong Kong University of Science and Technology, Clear Water Bay, Kowloon, Hong Kong 999077, China. [5]State Key Laboratory for Mechanical Behavior of Materials, Xi'an Jiaotong University, 710049 Xi'an, China. [6]Department of Physics and Organic and Carbon Electronics Laboratories (ORaCEL), North Carolina State University, Raleigh, NC 27695, USA. [7]College of New Materials and New Energies, Shenzhen Technology University, 518118 Shenzhen, Guangdong, China. [8]State Key Laboratory for Modification of Chemical Fibers and Polymer Materials, College of Materials Science and Engineering, Donghua University, 201620 Shanghai, China. [9]Institute of Polymer Optoelectronic Materials and Devices, State Key Laboratory of Luminescent Materials and Devices, South China University of Technology, 510640 Guangzhou, Guangdong Province, China. [10]These authors contributed equally: Siwei Luo, Chao Li, Jianquan Zhang, Xinhui Zou. ✉e-mail: sunym@buaa.edu.cn; hyan@ust.hk

