## [Peer Review File · Nature Communications]

Auxiliary sequential deposition enables 19%-Efficiency Organic Solar Cells processed from Halogen-free SolventsREVIEWER COMMENTS

Reviewer #1 (Remarks to the Author):

Remarks to the Author:

In this paper, the authors report a novel fabrication method for organic photovoltaic devices, which introduces the high crystalline additive DTT as a sequential processing auxiliary (SPA) in the conventional sequential deposition (SD) method to induce the formation of an ideal donor crystalline fiber network. The authors claimed that this Auxiliary Sequential Deposition (ASD)-LbL method achieves better vertical phase separation while effectively restraining charge recombination.

The device performance based on ASD-LbL reported in this manuscript exceeds that of traditional BHJ and SD methods by halogen-free solvents. In the part of thin-film morphology characterization, the results of GIWAXS, RSoXS, and AFM indicate that ASD-LbL is far better than traditional BHJ and SD methods in terms of obtaining ideal morphology. Also, according to the results of photophysical characterization, ASD-LbL devices exhibit suppressed recombination, efficient charge transfer process, and better charge extraction. These results evidentially support the author's conclusions, so we believe this manuscript is logical and straightforward.

Besides, the photovoltaic device based on halogen-free ASD-LbL achieved over 19% PCE. While the ASD-LbL devices show better stability than the traditional processing method. The breakthrough brought by this novel fabricating method is exciting. It breaks the limitation of using halogen-free solvents in traditional BHJ and SD methods. which is of great significance for the industrialization of organic photovoltaics.

Consequently, the author has made an innovative study on the organic photovoltaic devices fabrication method, and the reviewer recommends the publication of this paper after following minor revisions:

1. We noticed that in the manuscript and supplementary information, the authors provide detailed optimization data of the device, which contributes to the repeatability of the experiment. However, the batches of polymer can exert significant impact on the resulting photovoltaic performance. Therefore, in order to further ensure the repeatability of photovoltaic data, the molecular weight and PDI of the D18-Cl should be added in this manuscript.
2. In the introduction part, the author emphasizes the importance of this new processing method for OPV industry and commercialization, but only two small molecule acceptors L8-BO and L8-BOX are

measured in the paper, so the universality of this method is still in doubt. Therefore, we furtherly suggest the author to fabricate devices from other conventional small molecules such as BTP-4F and BTP-eC9 by ASD-LbL method to verify its universal.

3. Given the importance of industrialization, it is suggested that the author provide data on large-area devices and air-processing devices. This will help reviewers to further understand the commercialization potential of the ASD-LbL method.

4. In page 4, the authors mentioned that the sequential processing auxiliary (SPA)-the highly crystalline DTT can induce the preformation of a nano-porous polymer network from halogen-free solvent. As there are many newly developed solid additives used to fabricate high-efficiency OSCs. do the authors attempt other solid additives as SPA? is there any screening principles and criteria that a solid additive can be used as an efficient SPA?

5. In page 15, "Upon selective excitation on the L8-BO and L8-BO-X at 800 nm, we see the rise of a positive $\Delta T/T$ signal at ~ 830 nm (GSB) and a negative signal ~ 900 nm (ESA) at the early time ". Does the author mean the blend sample? Please specify.

6. In the same page, "both SD sample shows significantly longer τ_1 and τ_2 than the BHJ sample, which is due to the longer dissociation time". According to the previous explanation, it should be due to both dissociation and diffusion.

When exciting the acceptor exclusively, The SD sample shows significantly longer τ_1 and τ_2 . However, in Figure 6b & c, why the SD sample shows higher amplitudes than the BC?

Reviewer #2 (Remarks to the Author):

H. Zhang et al. reported a high efficiency OSC processed from halogen-free solvents by auxiliary sequential deposition. This work achieved 19% efficiency, but the method of device preparation lacks innovation (the method has been reported in [Sol. RRL 2022, 6, 2101096]), and the distribution and working mechanism of DTT are not clear. The reviewer therefore considered the work insufficient for publication in Nature Communications. More comments are given below, which hopefully will help to optimize the paper.

1. How is DTT distributed in the active layer, how does it affect the crystallization and aggregation of the donor and acceptor, and is it in the form of crystalline nucleus?
2. The authors need to discuss the mechanism by which the introduction of DTT acts on the VOC increase of the device.
3. For the absorption spectra of the mixed films, the reviewer considered it more reasonable to compare the unnormalized absorption spectra. This is because the crystallization of the material affects its absorption intensity, Especially, the EQE of the devices with different processing processes is significantly different at the long wavelength band.
4. The authors should give the test condition for photostability in the manuscript.
5. In paper 10, the author stated that :“This is much larger than that of L8-BO (4.0 nm), corresponding to the (111) peaks as reported in the literature, which suggests L8-BO-X has better long-range order than L8-BO.” More explanation or references are required.
6. For GIWAXS, the reviewer was very curious about the peak at $q_{xy}=0.75\text{\AA}^{-1}$ of D18-Cl:L8-BO-X ASD curve in OOP direction, to which material does it belong? What causes the multiple peaks between 1.5 \AA^{-1} and 2 \AA^{-1} and why are they not present in BC films? These discussions would be helpful to understand the role of DTT.
7. For FLAS, why does the relative intensity of the donor and acceptor peak differ significantly from the absorption spectra of the mixed film(Figure S3)?In addition, it is recommended that the authors discuss the effect of ASD on the time to crystallization of the donor and acceptor.

Point-to-point Response

(The original comments are marked in black, and the response are marked in purple)

Reviewer #1 (Remarks to the Author):

In this paper, the authors report a novel fabrication method for organic photovoltaic devices, which introduces the high crystalline additive DTT as a sequential processing auxiliary (SPA) in the conventional sequential deposition (SD) method to induce the formation of an ideal donor crystalline fiber network. The authors claimed that this Auxiliary Sequential Deposition (ASD)-LbL method achieves better vertical phase separation while effectively restraining charge recombination.

The device performance based on ASD-LbL reported in this manuscript exceeds that of traditional BHJ and SD methods by halogen-free solvents. In the part of thin-film morphology characterization, the results of GIWAXS, RSoXS, and AFM indicate that ASD-LbL is far better than traditional BHJ and SD methods in terms of obtaining ideal morphology. Also, according to the results of photophysical characterization, ASD-LbL devices exhibit suppressed recombination, efficient charge transfer process, and better charge extraction. These results evidentially support the author's conclusions, so we believe this manuscript is logical and straightforward.

Besides, the photovoltaic device based on halogen-free ASD-LbL achieved over 19% PCE. While the ASD-LbL devices show better stability than the traditional processing method. The breakthrough brought by this novel fabricating method is exciting. It breaks the limitation of using halogen-free solvents in traditional BHJ and SD methods, which is of great significance for the industrialization of organic photovoltaics.

Consequently, the author has made an innovative study on the organic photovoltaic devices fabrication method, and the reviewer recommends the publication of this paper after following minor revisions:

Reply: We appreciate the reviewer for the very positive evaluation and constructive comments on our work. According to the comments of the reviewers, we have made the following supplements to the manuscript.

Comment: We noticed that in the manuscript and supplementary information, the authors provide detailed optimization data of the device, which contributes to the repeatability of the experiment. However, the batches of polymer can exert significant impact on the resulting photovoltaic performance. Therefore, in order to further ensure

the repeatability of photovoltaic data, the molecular weight and PDI of the D18-Cl should be added in this manuscript.

Reply: Thanks for the suggestion. We have added the molecular weight and PDI details of the polymer donor to the Supplementary Information.

Table S12. The molecular weight and polydispersity index (PDI) of D18-Cl used in this work.

	M_p	M_n	M_w	M_v	M_z	M_{z+1}	PDI
D18-Cl	129903	52478	142214	249434	268913	414120	2.71

Comment: In the introduction part, the author emphasizes the importance of this new processing method for OPV industry and commercialization, but only two small molecule acceptors L8-BO and L8-BOX are measured in the paper, so the universality of this method is still in doubt. Therefore, we furtherly suggest the author to fabricate devices from other conventional small molecules such as BTP-4F and BTP-eC9 by ASD-LbL method to verify its universal.

Reply: Thanks for the comments. We have added a universality test based on the ASD method to the Supplementary Information. The test uses D18-Cl as the donor and BTP-eC9, N3, IDIC, and FCC-Cl as the acceptors. Due to the high processing temperature of the materials in a halogen-free solvent, the active layer morphology by the BC method deteriorated, leading to poor device performance. In contrast, the ASD-based devices showed better performance compared with the BC method. The SD processing avoids the negative impact of halogen-free solvent on active layer morphology. Hence, the open-circuit voltage, short-circuit current density and fill factor of the ASD devices fabricated by different acceptors are all improved, consequently proving the universality of the ASD method.

Figure S12. Universality test of ASD method based on multiple acceptors of (a) BTP-eC9, (b) N3, (c) FCC-Cl, and (d) IDIC.

Table S9. Universality test data of the ASD devices based on multiple acceptors of BTP-eC9, N3, FCC-Cl, and IDIC.

	V_{OC} (V)	J_{SC} (mA/cm ²)	FF (%)	PCE (%)
D18-Cl:BTP-eC9 ASD	0.873	26.883	0.733	17.195
D18-Cl:BTP-eC9 BC	0.868	25.909	0.647	14.545
D18-Cl:N3 ASD	0.855	26.213	0.721	16.140
D18-Cl: N3 BC	0.854	25.802	0.646	14.251
D18-Cl:FCC-Cl ASD	1.112	16.635	0.678	12.546
D18-Cl: FCC-Cl BC	1.105	16.386	0.575	10.400
D18-Cl:IDIC ASD	1.001	18.251	0.665	12.148
D18-Cl: IDIC BC	0.991	17.909	0.634	11.254

Comment: Given the importance of industrialization, it is suggested that the author provide data on large-area devices and air-processing devices. This will help reviewers to further understand the commercialization potential of the ASD-LbL method.

Reply: Thanks for the comments. We have added the data on air processing and large-area devices to the Supplementary Information. The results show that the ASD method can obtain outstanding results based on air-processed and large-area devices, showing excellent industrialization value.

Figure S10. J - V curves of the ASD devices of D18-Cl:L8-BO and D18-Cl:L8-BO-X under ambient conditions.

Table S7. Device data of the ASD devices of D18-Cl:L8-BO and D18-Cl:L8-BO-X under ambient conditions.

	V_{oc} (V)	J_{sc} (mA/cm ²)	FF (%)	PCE (%)
D18-Cl:L8-BO	0.894	25.31	74.43	16.84
D18-Cl:L8-BO-X	0.883	25.08	77.09	17.07

Figure S11. J - V curves of the ASD 1cm^2 area devices of D18-Cl:L8-BO and D18-Cl:L8-BO-X.

Table S8. Large-area photovoltaic data of the ASD devices based on D18-Cl:L8-BO and D18-Cl:L8-BO-X.

	V_{oc} (V)	J_{sc} (mA/cm^2)	FF (%)	PCE (%)
D18-Cl:L8-BO 1cm^2	0.894	25.461	0.646	14.696
D18-Cl:L8-BO-X 1cm^2	0.886	26.328	0.653	15.227

Comment: In page 4, the authors mentioned that the sequential processing auxiliary(SPA)-the highly crystalline DTT can induce the preformation of a nanoporous polymer network from halogen-free solvent. As there are many newly developed solid additives used to fabricate high-efficiency OSCs. do the authors attempt other solid additives as SPA? is there any screening principles and criteria that a solid additive can be used as an efficient SPA?

Reply: Thanks for the constructive comments. We tried some of the previously reported solid additives, but the performance was unsatisfactory. Based on the discussion of this manuscript, the authors believe that SPA should be selected very carefully. The SPA materials should have a sizeable rigid molecular plane with high crystallinity, suitable interactions with polymer donors, and good solubility in non-halogenated solvents. These characteristics may induce the formation of a porous donor structure during the swelling and crystallization processes. On the other hand, the SPA materials are more inclined to volatile solid additives with higher phase transition temperatures. In contrast, the phase transition temperature and boiling point of most solid additives are relatively

low to ensure volatility, which is not conducive to the successful ASD processing. In addition, most of the current solid additives contain halogens, which is challenging to adapt to large-scale industrialization. So, the choice of SPA is more inclined to halogen-free solid additives, which is also the demand for future OPV industrialization.

Comment: In page 15, "Upon selective excitation on the L8-BO and L8-BOX at 800 nm, we see the rise of a positive $\Delta T/T$ signal at ~ 830 nm (GSB) and a negative signal ~ 900 nm (ESA) at the early time ". Does the author mean the blend sample? Please specify.

Reply: We thank the reviewer for pointing this out. We have revised our manuscript and specified the spectra of the blend samples instead of pure acceptors. Here is the revised version:

"Upon selective excitation on the D18-Cl:L8-BO and D18-Cl:L8-BO-X at 800 nm, we see the rise of a positive $\Delta T/T$ signal at ~ 830 nm (GSB) and a negative signal ~ 900 nm (ESA) at the early time stage."

Comment: In the same page, "both SD sample shows significantly longer τ_1 and τ_2 than the BHJ sample, which is due to the longer dissociation time". According to the previous explanation, it should be due to both dissociation and diffusion. When exciting the acceptor exclusively, The SD sample shows significantly longer τ_1 and τ_2 . However, in Figure 6b & c, why the SD sample shows higher amplitudes than the BC?

Reply: We appreciate the reviewer for the comment. The SD sample does show a slower hole transfer than BC and ASD samples for both L8-BO and L8-BO-X blends. We think the higher amplitude of the donor GSB in the transient absorption spectra of SD samples may be due to the different optimized thicknesses of the SD and BC samples. The different donor distribution in the active layer results in a stronger donor GSB amplitude. Moreover, the less recombination of the SD sample than the BC one indicated by the TPV (as shown in Figure 5e) will also lead to the higher hole polaron concentration in the donor domain under the same excitation condition, which causes the higher donor GSB after hole transfer process (>100 ps).

Reviewer #2 (Remarks to the Author):

H. Zhang et al. reported a high efficiency OSC processed from halogen-free solvents by auxiliary sequential deposition. This work achieved 19% efficiency, but the method

of device preparation lacks innovation (the method has been reported in [Sol. RRL 2022, 6, 2101096]), and the distribution and working mechanism of DTT are not clear. The reviewer therefore considered the work insufficient for publication in Nature Communications. More comments are given below, which hopefully will help to optimize the paper.

Reply: We sincerely thank the reviewer for the comments and suggestions on this manuscript. We have noticed the publication mentioned in the comments (*Sol. RRL* 2022, 6, 2101096). This article doped the polymer donor by sequential processing method and slightly improved the device performance. However, this article significantly differs from our manuscript on the theoretical basis, research direction, and processing methods.

Firstly, in the previous publication, three different dopants were used to dope the polymer donor layer by sequential processing to enhance the device performance. Using sequential processing methods aims to avoid morphology damage led by the crystallization of dopants. In contrast, the high crystallinity of SPA is exploited to form a porous donor fiber network in our manuscript. Also, the SPA induces the planar stacking and high crystalline acceptor domain, which greatly optimizes the morphology and achieves high performance under halogen-free solvent processing. Thus, ASD breaks the limits of using halogen-free solvents in the conventional BC processing method. Therefore, the previous publication significantly differs from our manuscript regarding theoretical basis and research direction.

Secondly, in the previously reported articles, dopants eventually reside at the D/A interface of the active layer and modify the interface, which forms a multi-component blend system of the dopants and the active layer. In our manuscript, DTT acts as SPA, penetrating donor film during the swelling process and inducing the formation of a porous fiber network structure. Then, during the deposition of the acceptor, DTT is removed due to solvent washing and higher processing temperatures (we add an FTIR measurement in Supporting Information), so the active layer is still a donor/acceptor binary system. Therefore, there are also significant differences in the processing methods.

In addition, in our manuscript, we designed a new small molecule acceptor L8-BO-X with γ -branch sidechain to achieve a record efficiency in OSC processed by halogen-free solvent using the ASD method, proving the adaptability of this strategy to this processing mode. Also, in the subsequent revision, we added a universality test to confirm the universality of the ASD method to various acceptors such as BTP-eC9, N3,

IDIC, and FCC-Cl. Therefore, we believe that the previously published article does not affect the originality of our study.

Table R1. Summary of key differences between the literature and our work.

	Sol. RRL 2022, 6, 2101096	This work
Materials	PM6, Y6	D18-Cl, L8-BO, L8-BO-X (new material)
Mechanisms	Doping (Dopant remained)	Solid additives (Additives removed)
Processing solvents	Chloroform (Halogenated)	Toluene (Non-halogenated)
Solvent additives	1-chloronaphthalene	None
Performance	16.47%	19.04%

Comment: How is DTT distributed in the active layer, how does it affect the crystallization and aggregation of the donor and acceptor, and is it in the form of crystalline nucleus?

Reply: Thanks for the comments. As a volatile solid additive (*Advanced Materials*, 2021, 33, 2105301), the DTT was completely removed during the ASD processing. To verify this point, we performed a Fourier Transform Infrared (FTIR) Spectrometer measurement. According to the test results in **Figure S22**, the signal of DTT is concentrated in the wavenumber ranging from 500 to 1000 cm^{-1} . The signal can be observed in both DTT neat film and the reference film of BC method (with 100 wt% DTT as a solid additive in chloroform solvent, low-temperature deposition). Meanwhile, the signal is eliminated in the SD and ASD films, which proves that the DTT is removed during processing. On the other hand, **Figure S21** shows the changes in 2D GIWAXS signal. After rinsed by toluene, the signal belonging to DTT disappeared, which proves that DTT did not remain in the active layer during the ASD processing. During the ASD processing, the role of DTT in this method is to promote the formation of a donor fiber network during the crystallization process following the swelling effect and also induce the planarization stacking. Furthermore, it improves the crystallinity of the acceptor.

Figure S22. FTIR spectrum of DTT and D18-Cl:L8-BO-X blend films from BC (with 100 wt% DTT), SD, and ASD methods.

Figure S21. GIWAXS and AFM of (a) D18-Cl, (b) D18-Cl treated by spin-coating DTT solution on top, and (c) D18-Cl treated by spin-coating DTT solution and then toluene for rinse.

Comment: The authors need to discuss the mechanism by which the introduction of DTT acts on the V_{OC} increase of the device.

Reply: We thank the reviewer for the comment. The increased V_{OC} with the introduction of DTT comes from lower energy loss, which is evidenced by suppressed recombination of the ASD device. The suppressed recombination is indicated by the light-intensity-dependent V_{OC}/J_{SC} , fs-TAS, FLAS, and TPV in the manuscript.

In the ASD device, the casting of the NFAs onto the nanoporous D18-Cl film should result in better phase separation between the donor and acceptor, which significantly enlarges the D/A interfaces for efficient charge separation, highly crystalline domains for improving charge transport, and reduced charge recombination.

Due to the fine-tuned morphology, the ASD films exhibit the fastest exciton diffusion and dissociation in fs-TAS measurements (**Figure 6b** and **6c**), which will lead to reduced recombination. Furthermore, as discussed in the manuscript, the longer-lived acceptor GSB signals for the ASD films shown in **Figure 6d** and **6e** also indicates the suppressed recombination in the ASD sample.

Meanwhile, FLAS results indicate that L8-BO-X from the ASD film has a minimal peak shift among the three films, suggesting that the LUMO level of L8-BO-X from the ASD film is mostly invariable at different film depths. The minimal spatial fluctuation of energetic levels can effectively avoid the formation of low-energy localized states. Minimal spatial fluctuation of energetic levels can effectively prevent the construction of the low-energy localized states as traps along the charge-transporting pathways, which is beneficial for suppressed recombination.

Overall, the optimized morphology results in suppressed recombination, which is indicated in many characteristics as mentioned above..

Comment: For the absorption spectra of the mixed films, the reviewer considered it more reasonable to compare the unnormalized absorption spectra. This is because the crystallization of the material affects its absorption intensity, Especially, the EQE of the devices with different processing processes is significantly different at the long wavelength band.

Reply: Thanks for the reviewer's suggestion. We have added the unnormalized absorption spectra of the blend films in the Supplementary Information.

Figure S3. Absorption spectra of the blend films for (a) D18-Cl:L8-BO and (b) D18-Cl:L8-BO-X from the BC, SD and ASD methods.

Comment: The authors should give the test condition for photostability in the manuscript.

Reply: Thanks for the reviewer's suggestion. We have added the test environment in the manuscript.

Device photostability. Photostability of the devices were performed by using Maximum Power Point (MPP) tracking mode (YH-VMPP-IV-16). The encapsulated devices were measured under continuous illumination by LED light source (380-810 nm, one solar

intensity) under the ambient air conditions (the relative humidity was 40%, the tested temperature was 45 °C).

Comment: In paper 10, the author stated that :”This is much larger than that of L8-BO (4.0 nm), corresponding to the (111) peaks as reported in the literature, which suggests L8-BO-X has better long-range order than L8-BO.” More explanation or references are required.

Reply: We thank the reviewer for the very careful review of our manuscript. According to a previous publication (*Nature Energy*, 2021, 6, 605-613). The diffraction peak (11 $\bar{1}$) in the in-plane (q_{xy}) direction is the alkyl chain lamellar diffraction peak. Thus, we have revised the original description and also marked the reference source:

“L8-BO-X exhibits sharp peaks in the in-plane (IP) direction, especially for the one at $q_{xy} = 0.43 \text{ \AA}^{-1}$ showing a CL of 23.8 nm. This is much larger than that of L8-BO (4.0 nm), corresponding to the (11 $\bar{1}$) peaks as reported in the literature (*Nature Energy*, 2021, 6, 605-613), which suggests L8-BO-X has better long-range order than L8-BO.”

Comment: For GIWAXS, the reviewer was very curious about the peak at $q_{xy}=0.75 \text{ \AA}^{-1}$ of D18-Cl:L8-BO-X ASD curve in OOP direction, to which material does it belong? What causes the multiple peaks between 1.5 \AA^{-1} and 2 \AA^{-1} and why are they not present in BC films? These discussions would be helpful to understand the role of DTT.

Reply: Thanks for the comments. In the D18-Cl:L8-BO-X films processed from the ASD conditions, more diffraction peaks were observed on the GIWAXS patterns compared to D18-Cl:L8-BO-X BHJ and D18-Cl:L8-BO ASD. These new diffraction peaks do not originate from the donor and acceptor, as confirmed by the GIWAXS patterns of neat films. The reason for these results can be attributed to the new assembly stacking patterns generated by the interaction between the solid additive DTT and the non-fullerene acceptor L8-BO-X. During the spin coating of the L8-BO-X layer, DTT is redissolved in the toluene solvent of the acceptor solution and undergoes significant intermolecular interactions with L8-BO-X, which is then frozen in the final blend film. The GIWAXS results of neat films D18-Cl, D18-Cl:DTT, and D18-Cl:DTT-Tol further reconfirmed the above speculations. No new diffraction peaks were observed after toluene washing, only slight changes in peak intensity and broadening, indicating a change in donor crystallinity. This suggests that DTT does not form new intermolecular interactions with D18-Cl, allowing all DTT to be removed by the toluene solvent. The addition of DTT leads to a denser and larger crystal-scale ordered assembly stacking of L8-BO-X in the out-of-plane direction, which favors charge carrier transport in the

vertical direction.

Comment: For FLAS, why does the relative intensity of the donor and acceptor peak differ significantly from the absorption spectra of the mixed film(Figure S3)?In addition, it is recommended that the authors discuss the effect of ASD on the time to crystallization of the donor and acceptor.

Reply: Thanks to the reviewer's valuable comments. This phenomenon can also be observed in some previous publications (*Advanced Materials*, 2022, 34, 2204718 and *Advanced Materials*, 2022, 34, 2203379). We believe that this phenomenon is usually attributed to the aging of the active layer during the sending of the sample to the testing facility (the change of the absorption of the active layer over time in *Advanced Materials*, 2022, 34, 2203379, and the significant difference in the spectra of the FLAS and UV tests), as well as the experimental errors caused by systematic errors and additional test steps such as in Figures from *Advanced Materials*, 2022, 34, 2204718. Fortunately, this measurement aims to explore the relative differences in vertical phase separation between different processing methods, so the negative impact from aging, experimental, and systematic errors can be negligible.

Figures from *Advanced Materials*, 2022, 34, 2203379. UV absorption spectrum of blend films with different aging time and Film-depth-dependent light absorption spectra.

Figures from *Advanced Materials*, 2022, 34, 2204718. UV absorption spectrum of blend films and Film-depth-dependent light absorption spectra.

REVIEWERS' COMMENTS

Reviewer #1 (Remarks to the Author):

The quality of this paper is significantly improved to meet the high standard of Nature Communication. Therefore, I recommend this paper to publish on Nature Communication without revision.

Reviewer #2 (Remarks to the Author):

The author responded to all questions and the manuscript is acceptable.